

# Frost quakes in wetlands in northern Finland during extreme winter weather conditions and related hazard to urban infrastructure.

Nikita Afonin[1], Elena Kozlovskaya[1], Kari Moisio[1], Emma-Riikka Kokko[1], Jarkko Okkonen[2]

[1]Oulu Mining School, University of Oulu, Oulu, 90570. Finland

[2]Geological Survey of Finland, Espoo, 02151, Finland

*Correspondence to*: Nikita Afonin (nikita.afonin@oulu.fi)

**Abstract.** The paper reports the first results of experiment in northern Finland during winter, 2022-2023, that aimed to study seismic events caused by seasonal freezing in so-called Critical Zone (CZ) of the Earth. These events have attracted public attention recently, as multiple reports about them from local inhabitants in Arctic and sub-Arctic areas appeared recently in

social networks. To make an instrumental study of such events, to reveal relationship between their occurrence and winter weather conditions and to evaluate the possible hazard, we installed two high-resolution seismic arrays with co-located soil temperature stations at two sites in Finland, one of them being in the city of Oulu in the sub-Arctic area (65.04 $^0$N, 25.61 $^0$E) and the other one, above the Polar Circle in municipality of Sodankylä (67.36 $^0$N, 26.63 $^0$E). The equipment recorded continuous seismic and soil temperature data during November 2022-April 2023. Based on reports from the inhabitants of

Talvikangas (Oulu) about ground shaking and unusual noises on 6.1.2023 and their observations of new fractures on the roads there, we selected the time interval for identification of frost quakes originated during that day from continuous seismic records in Talvikangas and in Sodankylä. During the selected time interval, the extremely rapid air temperature drop of about -1.4 °C/hour in Talvikangas and -0.88 °C/hour in Sodankylä were observed. We identified and located two types of seismic events, namely, frost quakes with frequencies of about 10-20 Hz, with waveforms like those of tectonic events, and irregular-shape

frost tremors with frequencies of about 120-180 Hz. The sources of frost quakes in Talvikangas are mainly located on irrigated wetland while in Sodankylä about 50% of registered frost quakes were caused by ice fracturing on the Kitinen river. However, several relatively strong events with origin in wetlands were also recorded. A significant number of sources of frost tremors are confined to wetland areas cut by irrigation channels and to roads cleaned from snow during winter, both in Talvikangas and in Sodankylä. We calculated ground accelerations and ground velocities for strongest events from both groups and

compared them to equivalent properties of other seismic signals, like distant local earthquakes in the area, mining production blasts, cargo trains vibration. Our study shows that high-frequency frost tremors corresponding to surface fracture opening in the uppermost frozen surface layer of the thickness up to 5 cm can directly damage infrastructure objects like roads and basements of buildings. Surface waves, produced by frost quakes and propagating inside the shallow soil layer, have large enough ground accelerations at epicentral distances of hundreds of meters. Therefore, frost quakes should be considered as

phenomenon, that potentially can damage infrastructures and they have to be taken into account in seismic hazard assessment.



Our research is the first instrumental study of seismic events originated from wetland areas. These events occur as a result of interaction between the uppermost layer of the solid Earth's CZ and atmosphere processes that deserves further study.

## 1 Introduction

The Arctic and sub-Arctic regions are snow dominated cold environments. There are numerous consequences of climate change in these regions. The climate change impact analysis already indicates clear changes in wintertime conditions in these areas and predicts that these changes will be observed also in the future (IPCC, 2022). For example, the amount of precipitation and humid winter seasons will increase, and the amount of snow in the ground will become more variable in the Arctic regions. The processes initiated by climate change and resulting in extreme and unusual weather conditions are particularly influencing the critical zone (CZ) of the Earth, which includes the biological, chemical, physical, and geological materials and processes that work together at the surface region of the Earth (Brantley et al., 2007). The CZ stats from the upper limit of vegetation and continues all the way through the soils to the bottom of the ground water. Dynamic processes in the CZ are modulated by climate and weather conditions, hence these conditions have direct impact on such natural environment as wetlands, and urban infrastructures in direct contact with the CZ (roads, buildings foundations etc.) (Brantley et al., 2007; Giardino and Houser, 2015; Parsekian et al., 2015). To understand the effects of the climate change and to ensure sustainable and safe living environment it is essential to study dynamical processes in the CZ under influence of various weather conditions.

For Arctic and sub-Arctic regions, the predicted changes in the snow cover and the wintertime snow/rainfall relationship (IPCC, 2022) may change the dynamics of the freezing and thawing processes in the CZ. One of the manifestations of these changes are frost quakes that have been encountered recently in urban areas of several countries, such as Finland, Canada, and USA (Battaglia and Changnon, 2016; Leung et al., 2017; Okkonen et al., 2020), where they resulted in fracturing of roads and building foundations (Battaglia and Changnon, 2016; Okkonen et al., 2020). Although the number of instrumental observations of frost quakes is limited, such events and related damage are repeatedly reported in social networks by local people.

It has been demonstrated (Okkonen et al., 2020) that frost quakes can originate from the uppermost part of frozen soils. The process of fracturing is initiated by rapid decrease of air temperature. In combination with other factors, such as thin snow coverage and hydrogeological conditions in the uppermost subsurface, the mechanical properties of water-saturated upper soils are changing, resulting in release of the accumulated thermal stress by fracturing. It can be judged from this model that such events are probably not occurring solely in urban areas, but in other parts of the CZ as well. An essential part of the CZ are wetlands that are defined by having a water table near or above the land surface for sufficient time to cause the development of wetland soils (either mineral soils with redoximorphic features or organic soils with > 40 cm peat) and the presence of plant species with adaptations to wet environments (Canada Committee on Ecological (Biophysical) Land Classification et al., 1997; McKenzie et al., 2021). The mechanical fracturing in the wetlands can cause mechanical damage to vegetation (roots and collars of trees) and vegetated soils and even increase emissions of greenhouse gases. However, the severity of such fracturing and related hazard is not well-known due to the lack of instrumental seismic observations.



In this study we use continuous seismic data to investigate thermal stress release processes in upper soils and wetlands caused by unusual winter weather conditions in Arctic and sub-Arctic (Boreal) areas of Finland. To detect events, we analyzed the seismic data recorded by two high-resolution seismic arrays, each consisting of 45 short-period three component seismic stations, with co-located soil stations, in two geographically different sites in Northern Finland characterized by different weather, geological and hydrogeological conditions. We analyze the time series of weather parameters (air and soil temperature) to detect specific weather conditions favorable for origin of frost quakes. We use a specially developed algorithm to detect and locate the seismic events that could originate from the uppermost subsurface associated with the CZ. We report observations of two types of seismic events in the CZ during extreme winter weather event in January 2023 in these two sites and evaluate the related seismic hazard in terms of ground motion (ground acceleration and ground velocity). We discuss our results in the context of interaction between atmosphere processes and the solid Earth's CZ that need to be investigated in order to mitigate consequences of climate change.

## 2 Experiment description

Our experiment was conducted during winter 2022-2023, starting in early November and ending in late April. To record seismic events originating from the CZ, we selected two areas in Northern Finland with different weather conditions. The first site (Tähtelä) is located in the municipality of Sodankylä (67.36 $^0$N, 26.63 $^0$E). and the second one (Talvikangas) in the city of Oulu (65.04 $^0$N, 25.61 $^0$E) (Fig. 1).

The Oulu site is characterized by subarctic continental climate with average winter temperature (from November to March) of about -6.5 $^0$C (Weatherbase.com) and average snow depth of about 0.5 m (Rankinen et al., 2004). The second site located at the territory of Sodankylä Geophysical Observatory and characterized by Arctic climate conditions, with freezing, long, extremely snowy winters. The average snow depth during wintertime is about 0.9 m (Rankinen et al., 2004). The typical average winter temperature for this area is about -10 $^0$C (weatherbase.com). A common feature of both sites is that they are in the close vicinity of wetlands cut by drainage channels (Fig. 1 (b)). In our study we do not distinguish between different types of wetlands (bogs, fens, and marshes), as all of them are present in selected research areas.



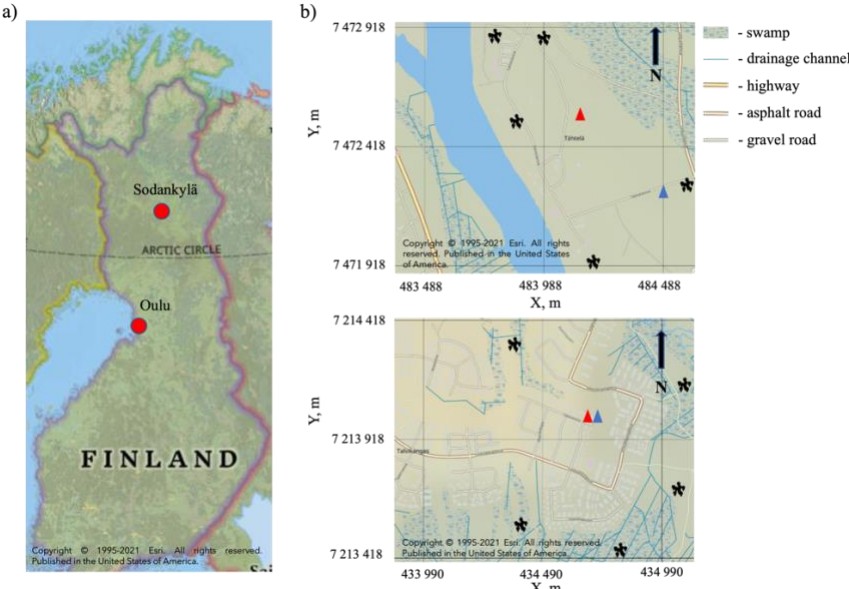

**Figure 1: (a) Geographical position of two sites of experiment in Northern Finland during winter, 2022-2023. (b) Installations of seismic arrays and soil stations in Tähtelä (upper panel) and Talvikangas (lower panel) blue triangles show soil stations; red triangles show broadband seismic stations; black dots indicate arrays of short period seismic stations. Coordinate system is**
**EUREF_FIN_TM35FIN. (Mention that units are in metres, X is Easting and Y is Northing)**

At the Oulu site the frost quakes were felt by local inhabitants earlier in January 2016 and January 2021. They reported in social network about ground shaking, cracks on the roads and unusual noises. The swarm of frost quakes on 6.1.2016 was investigated in Okkonen et al. (2020). It was demonstrated that the swarm of events occurred during rapid decrease of air
temperature from -15 to -25 °C in 24 hours (Fig. 2).

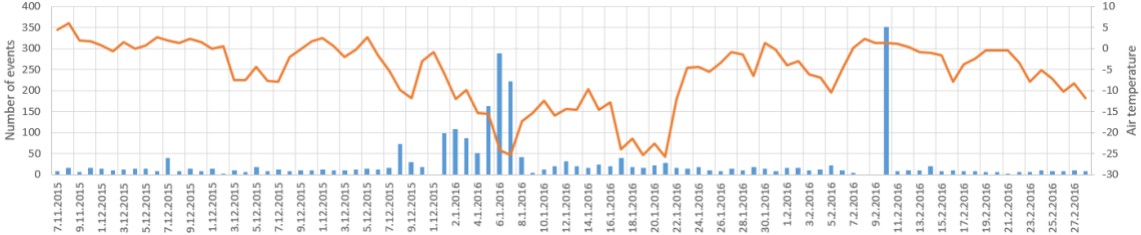

**Figure 2: Air temperature and number of seismic events recorded by permanent seismic station Oulu (OUL) during the winter 2015-2016 (Afonin and Kozlovskaya, 2019)**


However, the proper locating of these events was not possible, as the swarm was recorded only by a single permanent seismic station OUL. Thus, the main targets of our experiment were: 1) obtain instrumental recordings of frost quakes; 2) locate them





properly and 3) obtain additional evidence of connection between process of freezing and fracturing in the uppermost soils and weather conditions.

Because of the latter target, the following equipment was used in both sites. We installed data loggers CR10X, by Campbell Scientific Inc. (USA), which were continuously recording soil temperatures and soil moisture content at several depths between 10 and 50 cm in Talvikangas and between 10 and 80 cm in Tähtelä. The interval between sensors in each site was typically about 10 cm. These stations were recording data at the sampling interval of one hour from the end of October, 2022 to the end of May, 2023.

The seismological equipment included two broadband three component seismometers Guralp 3-ESPC Compact by Guralp Systems Ltd., UK (one per study site), and 90 (45 per site) short period three component seismic stations installed in 10 arrays (five per site) equipped by GS-ONE LF three-component geophones with GSX autonomous data recorders, by Geospace Technologies Ltd. (USA). All these instruments were obtained from the FLEX-EPOS pool of seismic instruments.

The configuration of arrays was selected considering frequency band of target signal and recommendations by Schweitzer et 115 al. (2012). The frequency band of 120-180 Hz was selected based on results of our earlier pilot experiment in Talvikangas during winter 2019-2020. In that experiment we recorded three component continuous seismic data by a single station in Talvikangas equipped by broadband Trillium compact seismometer, by Nanometrics Inc. (USA). The results of our pilot frost quake experiment (Okkonen et al., 2020) suggest that the seismic signal excited by fracturing in the uppermost soils would have mainly horizontal polarization, as the main fracturing mechanism in that case is vertical fracture opening. Therefore, to 120 find the frequency band of signals caused by such a process, we calculated a time series of horizontal-to-vertical spectral ratio (HVSR) from our seismic data and compared it to the air temperature variations (Fig. 3 (b)). The HVSR maxima at frequency of about 150 Hz is present when the air temperature is below zero and disappears when the temperature is above zero. Based on this observation, we suggest that the HVSR maxima at these frequencies may be caused by appearance of multiple small-scale fractures in the uppermost soil during its freezing in the beginning of winter season. To record the signals caused by 125 fracturing in this specific frequency band, we designed a particular seismic array configuration (Fig. 3 (a)) for short-period instruments.





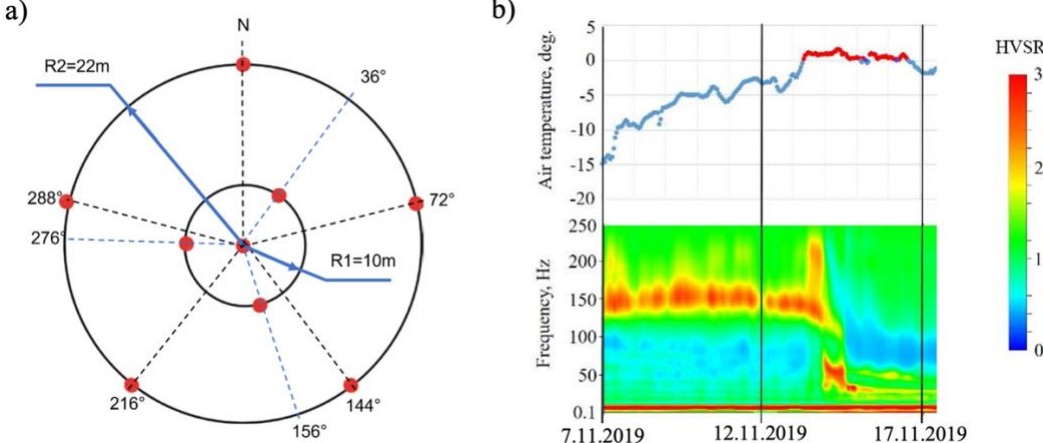

**Figure 3: Array configuration (according to recommendation by Schweitzer et al. (2012)): a) configuration of the array; b) HVSR,**
**calculated from continuous seismic data, recorded in Talvikangas in November 2019 compared to air temperature.**

The inner ring has a radius of 10 m and consists of 3 seismometers. The outer ring has a radius of 22 m and consists of 5

seismometers. All of seismic sensors (Fig. 3 (a)) were oriented according to ZNE coordinate system with Z pointing down.

The radii were chosen based on the assumed wavelengths of the target signal, which were estimated from the frequency band

and seismic wave velocities in upper soils. Figure 4 (a) shows the 1-D seismic velocity model for the Talvikangas obtained

from results of controlled-source refraction seismics near the site of broadband temporary seismic station (Fig. 1) confirmed

by results of co-located GPR survey that shows approximate thickness of sedimentary layer there (Fig. 4 (b)). According to

the data of soil types from the Geological Survey of Finland (GTK) and visual observations, the seismic equipment at both

sites were installed on similar sandy sediments. Thus, the velocity model (Fig. 4 (a)) is appropriate for event locating at both

sites.

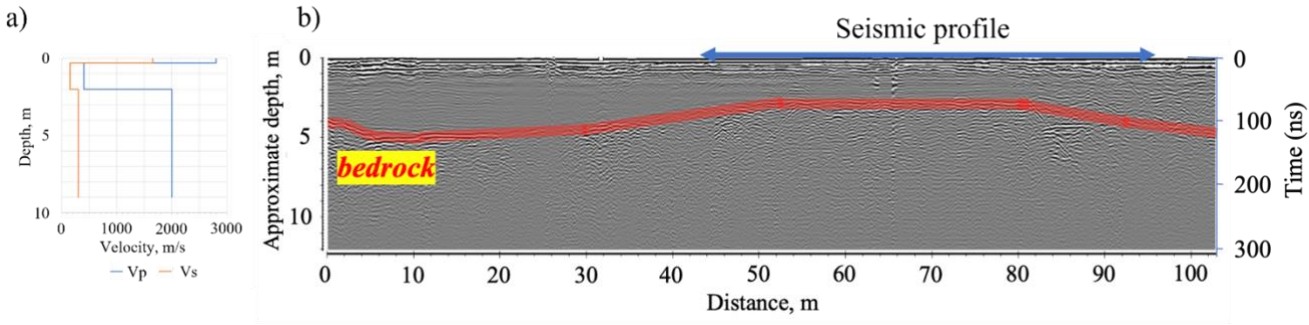


**Figure 4: (a) Near-surface 1-D velocity model of Talvikangas district of Oulu with the uppermost frozen layer of 30 cm added. (b)**
**results of GPR survey.**

As the velocity model was obtained from the refraction profile data recorded on unfrozen soils, we added one additional upper

layer of 30 cm thickness representing the frozen water-saturated soil. The layer thickness was constrained also by the data of





soil station in Tähtelä that shows the largest thickness of frozen layer. The velocities of this layer correspond to those of frozen sands saturated with water (Dortman, 1992).

The seismic instruments recorded seismic data from October 2022 until April 2023 with sampling frequency of 500 Hz. Generally, the data acquisition was designed in such a way that seismic sensors were not replaced, but 90 seismic short-period sensors and 90 GSX data loggers were simultaneously in the field, while the other 90 data loggers were under service. The

regular service included battery charging and raw data downloading from the data loggers. Such a scheme ensured continuous data recording by the arrays. The battery capacity of GSX loggers is enough for continuous recording lasting for approximately 40 days, hence they were serviced every moth. The service also included replacement of memory cards of broadband stations and batteries of soil stations.

**3 Air and soil temperature during winter 2022-2023**

The air and ground temperature time series during November 2022- January 2023 are shown in Figure 5 (a). Air temperatures are taken from Finnish meteorological institutes download service and stations Oulu (Pellonpää) and Sodankylä (Tähtelä) (Finnish meterorological institute open data, weather observations). The air temperatures were quite variable during the observation time, however, negative ground surface temperatures at both sites were reached by the end of November. After that the average soil temperatures at the depth of more than 10 cm in Talvikangas and more than 30 cm in Tähtelä remained

above zero, indicating the approximate average thickness of the initially frozen layer. Rapid air temperature drop occurred on both sites on 6.1.2023 (Fig. 5 (a), (b)) and, during that episode, the local inhabitants in Talvikangas heard unusual loud noises in the early morning. On the same day, they detected several new cracks on the roads and in walls of building foundations. In Talvikangas the air temperature dropped from -9.6 to -21 °C during 8 hours, which corresponds to -1.4 °C/hour gradient in average. In Tähtelä, the air temperature dropped from -9.1 to -34 °C during 28 hours, which corresponds to -0.88 °C/hour

gradient in average. In Tähtelä this was the lowest recorded air temperature as well as one of the highest negative gradients of the air temperature during winter 2022-2023.

Nevertheless, the soil temperature did not change such significantly on that day in Talvikangas, while in Tähtelä an abrupt temperature decrease in the uppermost 30 cm of the soil is seen on the next day data.

Considering this abrupt air temperature change and its similarity with the observations in our previous frost quake study

(Okkonen et al., 2020), and recognizing the observations of local inhabitants, it was reasonable to concentrate the seismic data analysis to detect and locate possible frost quakes on that specific day.



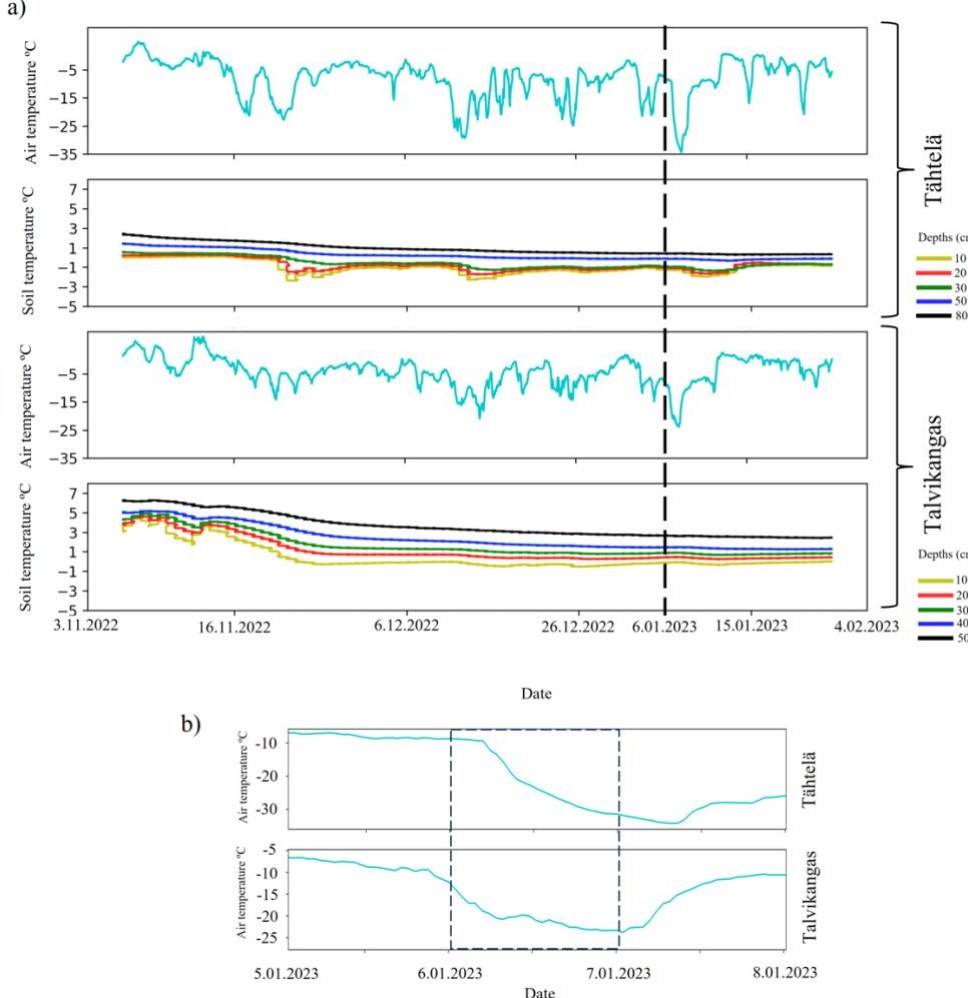

**Figure 5: Soil and air temperature observations during winter 2022-2023: a) time series for the whole observation time; b) time series on the beginning of January.**

## 4 Seismic data processing

The visual analysis of continuous recordings of both types of instruments showed that two types of waveforms can be recognized. The first type corresponds to the pulse-type events with clear onset of the signal and visible arrivals of compressional (P), shear (S) and surface waves (hereafter referred as events or frost quakes), and the second type corresponds to tremor-type signal with longer duration without clear onset (hereafter referred as frost tremors), but with characteristics different from waveforms produced by sources of anthropogenic origin (cars, snow cleaners etc.). As the considered area is aseismic in respect with tectonic earthquakes (Institute of seismology, University of Helsinki: https://www.helsinki.fi/en/institute-seismology), we can also exclude the tectonic origin of these waveforms. To our



knowledge and based on online real-time Bulletin of Seismic Events in Northern Europe (Institute of Seismology, University of Helsinki) no blasting activity was going on near our observation areas during that day. Hence, we assumed that both types of waveforms correspond to frost quakes or frost tremors, respectively. Thus, the seismic data processing consisted of two parts. Depending on the type of target event, we analyzed seismic data in two frequency bands: the first one is of 2-20 Hz and corresponds to frost quakes, while the second one with frequencies of 120-180 Hz corresponds to tremors. Both frequency bands were selected based on results of our previous pilot experiment during winter 2019-2020.

### 4.1 Detecting and locating frost quakes

For event detection we used seismograms recorded by the arrays that were bandpass filtered of 2-20 Hz and stacked using beamforming algorithm. An example recording of the strongest detected event (Z-component) and particle motion diagrams are shown in Fig. 6 (a) and Fig. 6 (b), respectively. In Fig. 6 (a) three distinct arrivals corresponding to P-, S- and surface waves are seen. The time difference between arrivals of P-and S-waves indicates that the event source is located at distances from about 300 m to 1100 m from the arrays. Despite some visible similarities with tectonic earthquakes or blasts, the recording of this event has some specific features. The P-wave is polarized mainly in the horizontal plane, meaning that its incidence angle is close to 90°. The S-wave is polarized in the vertical plane. At the same time, the particle motion diagram of the surface wave is typical for Rayleigh wave (R). These features suggest that the source is in the very shallow subsurface and the S-waves are propagating inside the shallow subsurface layer with apparent velocities of about 480 m/s through the "dry land" and about 430 trough the swamp. These velocities are not in contradiction with average velocity in the velocity model (Fig. 4 (a)).

We selected the high-quality recordings of events at both sites using visual analysis of stacked seismograms and picked both P- and S-wave arrivals. We used these arrival times and velocity model (Fig. 4 (a)) to calculate the locations of the event sources. The strongest event was in the wetland (Fig. 6 (c)).





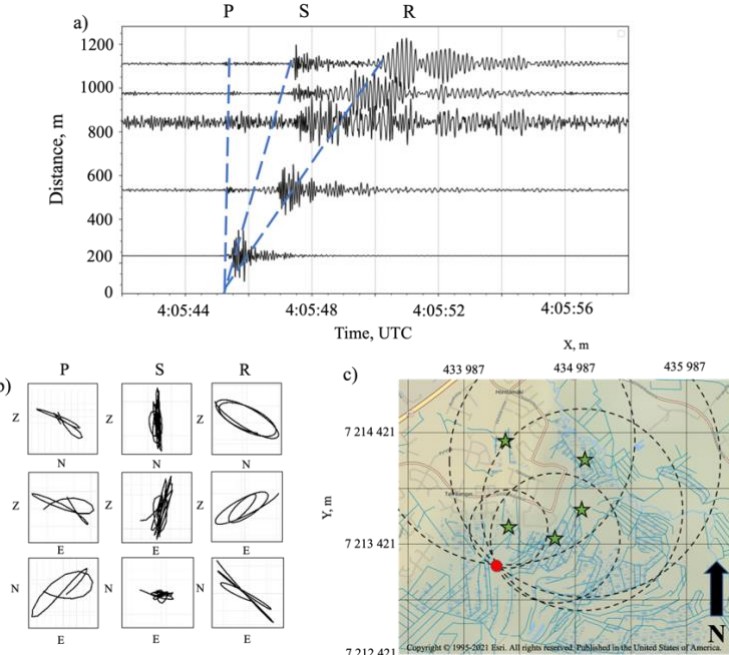

**Figure 6: The strongest frostquake, detected on 6.1.2023 in Talvikangas: a) vertical component seismograms; b) particle motion diagrams for P- S- and Rayleigh (R) waves; c) location of the epicenter indicated by red dot.**

The epicenters of detected and located frost quakes on 6.1.2023 at both sites are shown in Fig. 7. This includes 11 frost quakes in Talvikangas and 34 frost quakes in Sodankylä. Typically, the error of epicentres location is about 50-150 m.

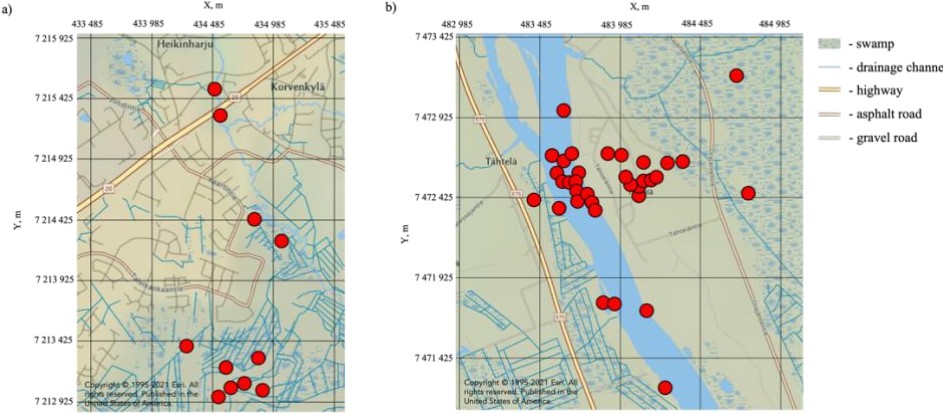

**Figure 7: Locations of the frost quakes, detected on 6.1. 2023: a) Talvikangas; b) Tähtelä.**

The frost quakes sources in Talvikangas are mainly located on wetlands (swamps) (Fig. 7 (a)). A small number of the frost quakes originated from a vicinity of the nearest highway, located close to a bridge over a small river, so are probably caused by ice fracturing. The site in Sodankylä had more frost quakes (Fig. 7 (b)) and about 50% of the registered frost quakes were



caused by ice fracturing on the Kitinen river. Significant part of frost quakes originates from small roads which are kept open
from snow during winter. Several events also originated from wetland areas.

We converted digital seismograms to ground velocities and ground accelerations, to evaluate possible hazard for roads and building foundations in the vicinity of the frost quakes. Seismograms and spectrograms for the above-mentioned strongest event in Talvikangas recorded by the seismic array with minimum epicentral distance about 300 m are shown in Fig. 8. The Rayleigh wave is characterized by relatively wide frequency band of approximately 5-30 Hz and the amplitude maxima are
seen in frequencies 15-20 Hz.

The spectrogram of the vertical component (Fig. 8 (b)) shows that the peak ground acceleration of Rayleigh wave corresponds to frequencies of about 20-25 Hz and it is about 0.9-1 m/s$^2$, while the peak ground velocity is about 0.0065-0.007 m/s.

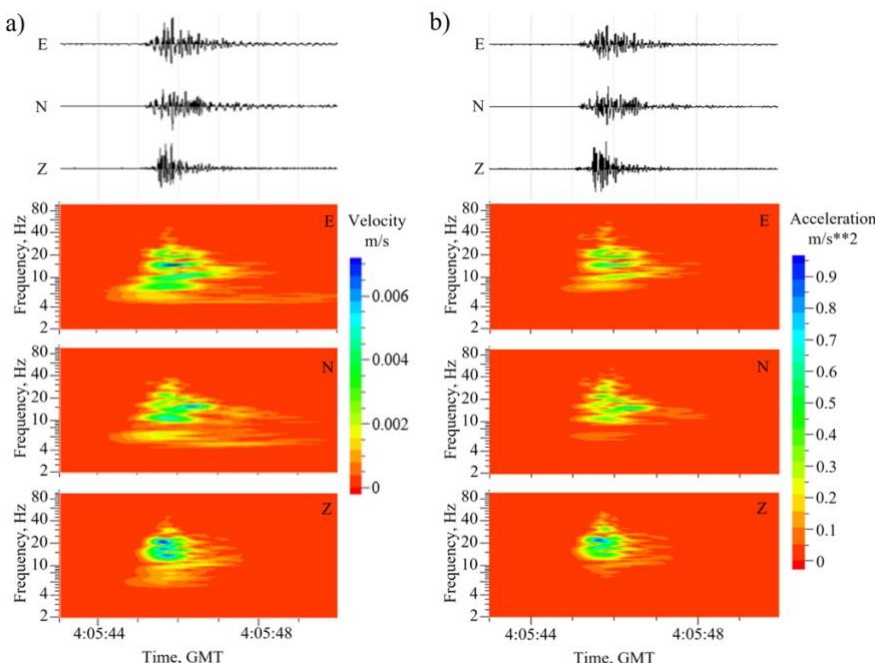

**Figure 8: Seismic records and corresponding spectrograms of the strongest frostquake in Talvikangas on 6.1.2023, recorded by the**
**closest seismic array to epicenter (epicentral distance is about 300 m): a) ground velocities; b) ground accelerations.**

### 4.2 Detecting and locating tremors

In this study we call "tremor" a small-scale seismic event that has the waveform detectable by the array, but without clear onset and without clear arrivals of body waves. We found such events already in our pilot experiment in 2019-2020, but their
origin and locations were not possible to define, as they were recorded only by a single seismic instrument. The frequency band of such events was 120-180 Hz, which corresponds to seismoacoustic emission phenomenon, namely, forming microcracks in the ground accompanied by release of microseismic energy (Goodman and Blake, 1965; Cadman et al., 1967).





During massive microcracking, such events appear as swarms, in which signals of individual events are often overlapping. The individual events in tremors usually look like impulses with duration of about 0.5-0.8 s. As these events are not strong, only surface waves can be recognized on seismograms (Fig. 9). Comparison of waveforms of such events with that of the single frost quake described in Okkonen et al. (2020) suggests the similar source mechanism, namely, vertical fracture opening in the very shallow frozen subsurface layer. However, it is not possible to use the same algorithm for tremor detection and locating as with frost quakes, because the tremors have no clear signal onset.

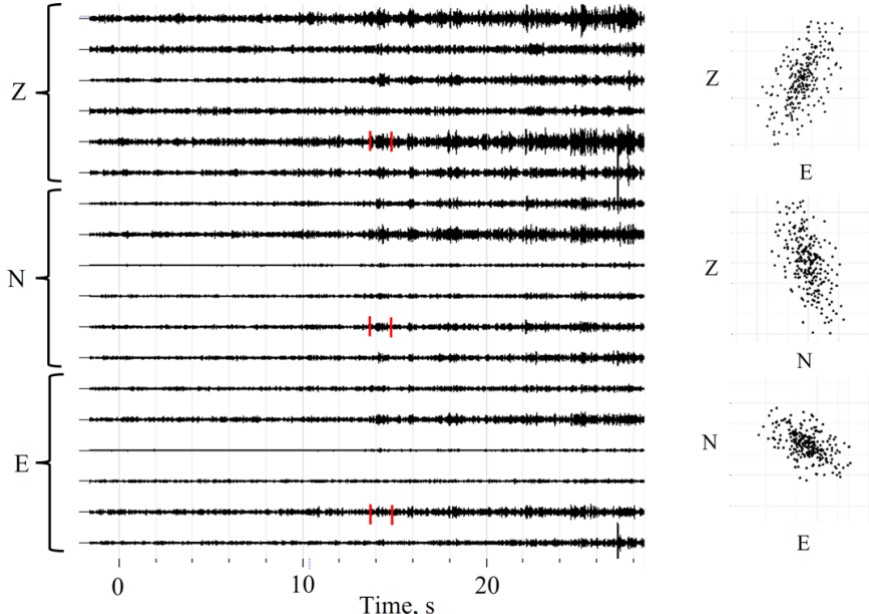

**Figure 9: An example of tremors swarm on frequency band of 120-180 Hz, recorded by our seismic arrays (a). Red lines indicate a single tremor signal recorded by the single array station. (b) particle motion diagrams of typical tremor showing elliptical polarization.**

To detect tremors from continuous seismic recordings and calculate their source locations, we used an algorithm shown in Fig. 10. The detection of tremors in this algorithm is based on cross-correlation of seismograms in time domain (interferometry) (Afonin, 2022). The calculation of source coordinates is done using direct grid search method (Tarantola, 1987), in which the coordinates of the trial source with minimum misfit between theoretical and measured travel times are considered as the true coordinates of the source.



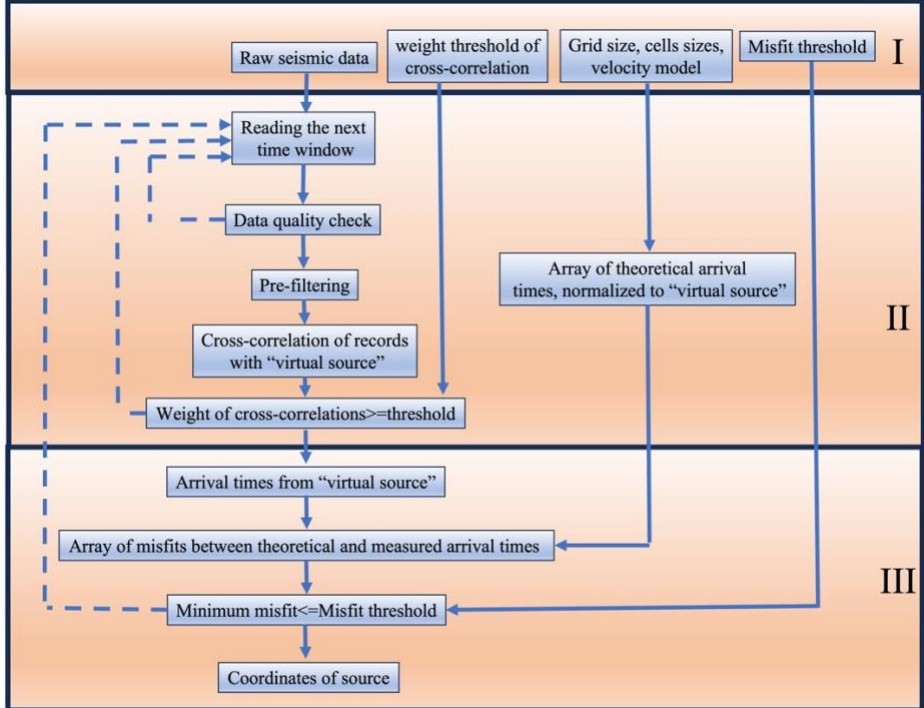


**Figure 10: Block-scheme of detection and location algorithm: I – setting processing parameters and loading input data in miniseed format; II – processing of seismograms; III – source locating.**

The algorithm includes three stages:

      1)    Setting processing parameters and uploading input data.

255         On this stage, the raw data, recorded by all stations of the array for considered time window are uploaded and the input parameters are set. The input parameters are 1) weight threshold of cross-correlation, the minimum average cross-correlation coefficient of seismic records, that initiated next steps of data processing; 2) grid size, the number of cells of grid, where each grid point is considered as potential source location in direct search algorithm; 3) cells size, step (in meters) in direct search algorithm; 4) velocity model of the ground, for calculating theoretical travel times between possible source

location and seismic receiver; 5) misfit threshold (the value of misfit between theoretical and measured travel times).

        In the case of low data quality as well as of low weight of cross-correlations, the algorithm does not initiate the third stage and proceeds to the beginning to read the next time window.

      2)    Processing of seismograms.

        This stage includes reading the data for the considered time window, data quality check, prefiltering the data by

bandpass filter and calculation of cross-correlation functions between seismogram of the selected "virtual source" (that is, one of the receivers of an array) and the other seismograms, recorded by all other receivers in the array. The weight of cross-correlation is calculated as an average of maximum values of all cross-correlation functions. If this value is less than the threshold, then several new virtual sources are tested. This is necessary to exclude the effect of rejection of an event because



of low quality of the virtual source seismic record. In the case when the obtained weight is larger than the threshold, the

algorithm initiates the next stage of processing. At the same time, an array of theoretical travel times from all grid points to

the virtual source position are calculated and normalized as:

$$t_{ij}^t = S_{ij}|\boldsymbol{r}_{ij}| - S_{i0}|\boldsymbol{r}_{i0}|, \tag{1}$$

where $S_{ij}$ is slowness along radius vector $\boldsymbol{r}_{ij}$ (Fig. 11).

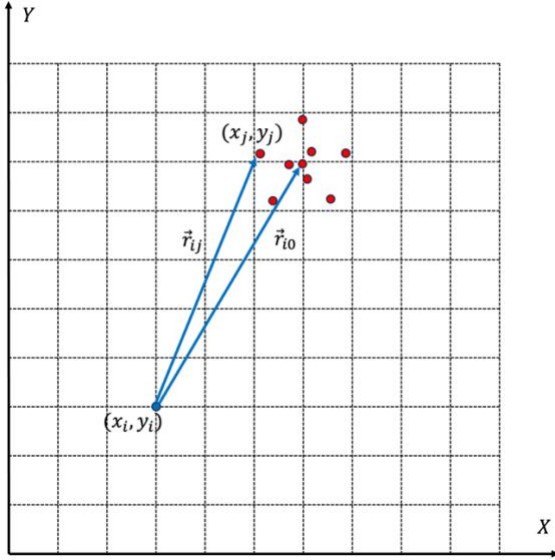

**Figure 11: Geometry illustrating direct search location algorithm: blue dot show source position been tested; the blue dot shows the source position being tested; the red dots show the locations of seismic sensors.**

    3)    Source location.

This stage calculates an array of misfits between theoretical travel times for each grid point and measured arrival times. In our

study, we are calculating misfit between observed and calculated travel times as (Fig. 11):

$$\sigma_i\big(\boldsymbol{d}^{obs}, \boldsymbol{d}_i^{calc}\big) = 100 \sum_{j=1}^n \big|(d_i^{obs} - d_{ij}^{calc})/d_{ij}^{calc}\big| \tag{2}$$

where $i$ is a grid point index; $j$ is receiver index; $\boldsymbol{d}^{obs}$ is a vector of observed travel times; $\boldsymbol{d}_i^{calc}$ is a vector of calculated travel

times from source, located in $i$-th grid point.

After this we find the grid point index that corresponds to the minimum misfit as:

$$k = argmin(\sigma_i\big(\boldsymbol{d}^{obs}, \boldsymbol{d}_i^{calc}\big)) \tag{3}$$

The minimum misfit $\sigma_k\big(\boldsymbol{d}^{obs}, \boldsymbol{d}_k^{calc}\big)$ is compared with the misfit threshold $\sigma_{thresh}$ and if $\sigma_k <= \sigma_{thresh}$, the grid point with

index $k$ is considered as the true source position and coordinates $(x_k, y_k)$ as the true source coordinates. Otherwise, the tremor

is considered as result of detection error.



The following inputs were used in the algorithm. Assuming the signals at these frequencies and energies usually attenuate very quickly, we considered an area of 1 x 1 km around the virtual source, that usually was the center sensor of the array. A vector pointing to this sensor from a grid point with coordinates $x_i$ and $y_i$ is $\boldsymbol{r}_{i0}$ (Fig. 11). The grid cell size (in our case 25x25 m) corresponds to the source size, that was estimated from frequency band of the target signal. Because we have only 1D velocity model for both sites, we assumed homogeneous near surface structure. As a cross-correlation weight threshold we used the value of 0.65. The misfit threshold was selected to equal 10% difference. The length of input time windows was 0.8 s, and the maximum time lag of cross-correlation function was 0.3 s. Such parameters of cross-correlation were selected considering the possible duration of tremor signals and the near-surface velocity model.

Applying this algorithm to the data recorded on 6.1.2023 at both sites, we detected and located 419 tremors in Talvikangas and 217 in Tähtelä. For these events we also calculated the density, that is, the number of events per grid cell raster (Fig. 12). In Talvikangas our results were partially confirmed by reports of local people about fractures on the roads that they noticed after the events on 6.1.2023. The tremors corresponding to these visible fractures were coincident with some of the epicenters (Fig. 12). Significant number of tremors sources are located in the wetland areas around Talvikangas. In Tähtelä a lot of tremor sources are on the river ice. The areas with the highest density of tremor sources do not correspond to the largest wetland in the East, although some tremors were located also there. Some of the sources are located in the wetland near the highway cut by irrigation channels and also to the roads that are kept open from snow during winter.

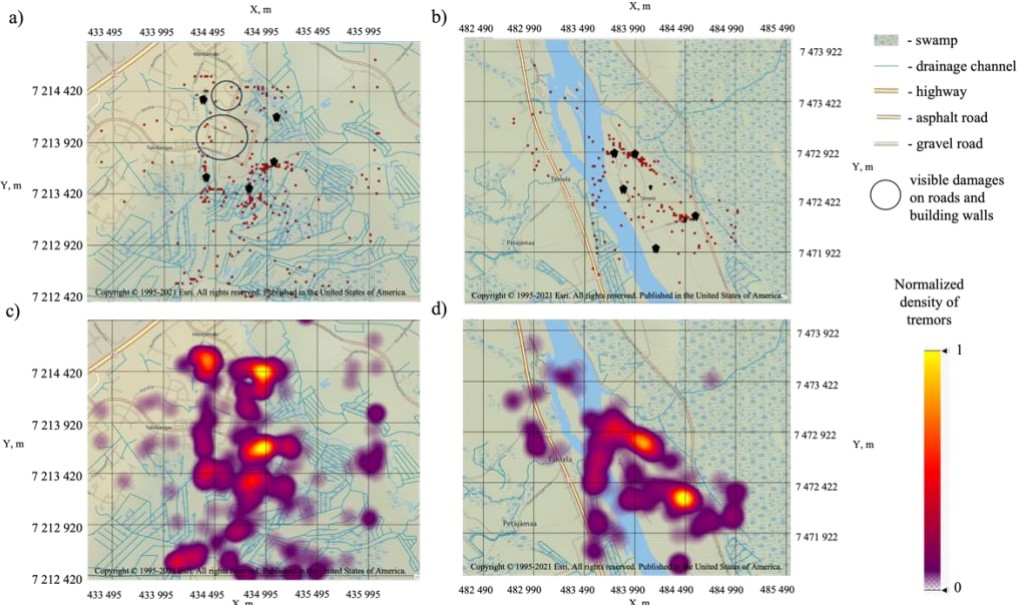

**Figure 12: Distribution of tremors sources on the map (red dots – sources of tremors, black dots – short period seismic stations): sources position in Talvikangas (a) and in Tähtelä (b); density of sources (number of sources per cell 25 x 25m), normalized to their maximum for considered area: (c) Talvikangas, (d) Tähtelä.**





## 5 Results and discussion

In our study we report weak seismic events induced by certain winter weather condition in Arctic and sub-Arctic areas, namely, frost quakes and frost tremors, that are usually not considered in traditional seismology. These events occur as swarms due to massive fracturing in the shallow subsurface layer, that is, inside the CZ. At the Talvikangas site we detected 11 frost quakes, mostly originating from irrigated wetland. At the Tähtelä site we detected and located 34 frost quakes, about half of them with origins on river ice, but some of them also occurred in the wetland area. We also detected and located sources of high-frequency

(120-180 Hz) frost tremors in both study sites. Significant number of tremors originated from wetland areas around Talvikangas, while a lot of sources were on river ice in Tähtelä. Some sources of tremors in Talvikangas are confined to roads or located close to buildings and some of these locations were confirmed by local inhabitants, who heard cracking noises and noticed new cracks on the roads.

Our results suggest that the events swarms were initiated by regional-scale processes in the atmosphere that resulted in rapid

air temperature drop to -20 °C in Talvikangas and to -30 °C in Tähtelä on 6.1.2023. In Talvikangas the air temperature dropped from -9.6 to -21 °C during 8 hours, which corresponds to -1.4 °C/hour in average. In Sodankylä the air temperature dropped from -9.1 to -34 °C during 28 hours, which corresponds to -0.88 °C/hour in average. Noticeably, the temperature did not reach extreme cold values for Arctic and sub-Arctic regions. According to observations of the air temperatures in Finland by Finnish Meteorological Institute (FMI), the coldest temperature of -51.5 °C was recorded in Finland during winter, 1999 in Kittilä,

which is located approximately at the same latitude, as Sodankylä (Finnish meterorological institute open data, weather observations). In Talvikangas on 6.1.2016 (Okkonen et al., 2020) similar rapid air temperature decrease to -20 °C was observed. Similar temperature phenomenon occurred in January 2021, resulting in fracturing of roads around city of Oulu (including Talvikangas), which were reported in social networks by local people. A common feature is that all three episodes of massive fracturing in Talvikangas in January 2016, 2021, and 2023 and in Tähtelä in January 2023 occurred when the stable negative

ground temperature was reached by the end of seasonal freezing period. However, the main factor that initiated the fracturing appears to be namely the rapid air temperature decrease, but not necessarily to the extreme low values (Fig. 13).

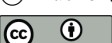



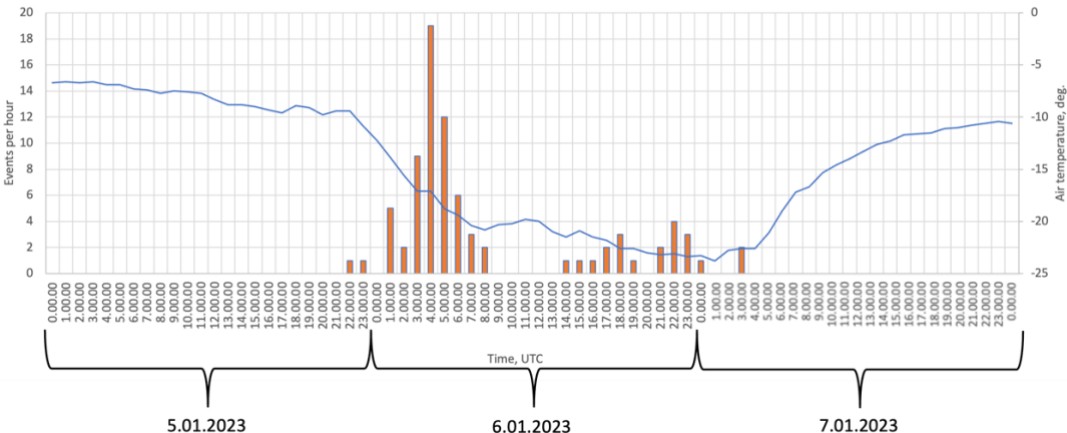

**Figure 13: Number of seismic events per hour, recorded by seismic array in Talvikangas 5.01.2023-7.01.2023.**


 This factor can be characterised by the value of air temperature time derivative (cooling rate). The similar relationship between rapid air temperature decrease and initiation of massive fracturing in ice wedges in permafrost in Spitsbergen was noticed also by O'Neill and Christiansen (2018). They observed that the fracturing in ice wedges occurs when the ground surface was cooling rapidly due to air cooling, at a rate of about -0.8 °C/hour, which is similar to the cooling rates observed in our study.

Information from local inhabitants in Talvikangas demonstrated that the frost quakes and tremors occurred during the same day when also damage to roads and buildings was noticed. Thus, it is important to evaluate qualitatively the possible hazard due to these events. This problem is not considered in the traditional earthquake seismology that deals mainly with tectonic and induced earthquakes and their hazard. One of the main parameters used to access the seismic hazard in seismology is event magnitude that is usually evaluated from far-field seismic recordings. As the frost quakes have sources of smaller size, they

could be studied using near-field instrumental recordings only. For evaluation of the seismic hazard caused by these events we calculated peak ground accelerations and peak ground velocities of the strongest frost quake recorded in Talvikangas and strongest frost tremors both in Talvikangas and Tähtelä and compared them to corresponding parameters of local earthquakes in Finland, underground production blast in Malmberget mine (Sweden), vibrations near the railroad and some large events recorded at teleseismic distances (Table 1).





Table 1 – Ground velocities and ground accelerations, produced by different seismic events.

| Event | Magnitude | Epicentral distance, km | Peak ground velocity, m/s | | Peak ground acceleration, m/s² | | Frequency, Hz |
|---|---|---|---|---|---|---|---|
| | | | Horizontal | Vertical | Horizontal | Vertical | |
| Frost quake in Talvikangas 6.01.2023 | - | 0.3 | $7*10^{-3}$ | $7*10^{-3}$ | $5.6*10^{-2}$ | $9.67*10^{-2}$ | 10-30 |
| Tremor in Talvikangas | - | 0.05-0.25 | $3*10^{-5}$ | $5*10^{-5}$ | $10^{-2}$ | $2*10^{-2}$ | 130-160 |
| Tremor in Sodankylä | - | 0.05-0.25 | $3*10^{-6}$ | $3*10^{-6}$ | $10^{-3}$ | $2*10^{-3}$ | 130-160 |
| Earthquakes in Northern Finland (Pavlenko and Kozlovskaya, 2018) | | | | | | | |
| 30.12.2009 | 2.2 | 137.5 | - | $4*10^{-6}$ | - | - | 2-18 |
| 15.03.2011 | 2.1 | 123 | - | $4*10^{-6}$ | - | - | |
| 18.01.2016 | 1.9 | 27 | - | - | - | $1.55*10^{-4}$ | |
| 19.03.2016 | 4.1 | 159.8 | - | - | - | $8.35*10^{-4}$ | |
| Underground blasts in LKAB Malmberget mine (Zhang, 2012) | - | 0.25÷0.3 | - | $(5÷19.5)*10^{-3}$ | - | - | 20-25 |
| Vibration, produced by cargo train (Antonovskaya et al., 2017) | - | 0.02 | - | - | $10^{-3}$ | $10^{-3}$ | 4.7-5.5 |
| Tohoku, Japan, 2011 (Santulin et al., 2014) | 9 | 131 | - | - | 2.75 | 1.92 | 1.5-4 |
| Imperial Valley, USA (Santulin et al., 2014) | 6.6 | 4 | - | - | 0.44 | 1.64 | 1.5-4 |

The amplitudes of both vertical and horizontal ground acceleration of the strongest frost quake are quite large, as they are amplified due to propagation inside the sedimentary layer. According to Modified Mercalli intensity scale by the United States Geological Survey (Wald at el., 1999), the ground velocities of the frost quake in Talvikangas located at distance of 300 m from buildings and roads correspond to II-III class (weak perceived shacking without potential damage). However, the ground accelerations produced by the considered frost quake, correspond to V class (moderate perceived shaking with very light potential damage), which agrees with the evidence of local inhabitants. As seen in Fig. 7, the largest amplitude of ground acceleration produced by the strongest frost quake is at frequencies about 10 Hz, which is similar to natural frequency of 1-store buildings defined by empirical "Rule of Thumb" (FEMA, 1998). This, in our opinion, explains appearance of fractures in foundations of some buildings in Talvikangas.

The strongest frost quake in Talvikangas produced vertical and horizontal ground accelerations that are two orders larger than the accelerations produced by regional earthquakes and about 10 times larger than amplitude of vibrations from a cargo train moving along the railroad located at the distance of 20 m. An interesting observation is that horizontal and vertical accelerations from high-frequency frost tremors located at distances of 50-250 m are comparable with those produced by cargo train. However, the frequencies of tremors are significantly higher than the frequencies of vibrations produced by the cargo trains.

The ground velocities produced by the strongest frost quake and registered at the distance of 300 m from the source are comparable with the velocities recorded at similar distance from a production blast in Malmgerget underground mine in Sweden. Moreover, the frequencies of Rayleigh waves in both cases are similar. As shown in Zhang (2012), such a level of



ground shaking was a source of concern for local inhabitants and required taking special measures (change in blasting technology) to decrease it. The ground velocities and ground accelerations produced by frost tremors are small, however, their appearance is the evidence of massive fracturing in the uppermost layer of frozen water-saturated soils and wetlands that are the part of CZ.

One interesting result of our study is the detection of unusual types of seismic events, namely frost quakes and frost tremors, originating from wetlands (Fig. 7 and Fig. 12). To our knowledge, instrumental observations of such events have not been reported previously. These events occurred mainly in irrigated wetlands cut by drainage channels. After autumn seasons these channels are filled with water that is frozen during winter and hence prone to fracturing. Such frozen channels in wetlands are comparable in size and in depths with ice wedges in permafrost areas. These wedges may experience massive fracturing related

to rapid air temperature cooling (O'Neill and Christiansen, 2018). This similarity may explain also fracturing in irrigated wetland revealed by our study, as it occurred during time interval with similar weather conditions. The origin of other events from non-irrigated wetland areas could be related to other natural structures capable to accumulate water, like marshes.

Our study shows that seismic events in wetlands in Arctic and sub-Arctic areas are capable to produce ground motions strong enough to damage the infrastructures, like roads and basements of buildings, located at distances of several hundreds of meters

from wetlands. In Arctic and sub-Arctic wetlands, fracturing can also cause mechanical damage to vegetation (roots and collars of trees), and also to create conditions for increasing greenhouse gases emission during wintertime. That is why this phenomenon deserves further studies.

**Conclusions**

The main results of our study can be formulated as follows:

1. Seismic experiment in northern Finland recorded continuous seismic and soil temperature data during November 2022-April 2023. The equipment was installed at two sites, namely, the urban Talvikangas area in city of Oulu (65.04 $^0$N, 25.61 $^0$E ) and the rural Tähtelä area in Sodankylä municipality (67.36 $^0$N, 26.63 $^0$E ). The study revealed seismic events (frost quakes) originating from shallow subsurface that is associated with the Critical Zone of the Earth.

    2. The detailed study of frost quakes swarms recorded at two experiment sites on 6.1.2023 showed that these swarms
400        occurred at the end of the period of seasonal freezing, when the stable negative ground temperature was established in the uppermost part of water-saturated soil. On the same day, the inhabitants in Talvikangas felt ground shaking and observed new fractures on the roads there.

    3. Analysis of continuous seismic data recorded 6.1.2023 at both sites allowed to identify and locate two types of seismic events, namely, frost quakes with frequencies of about 10-20 Hz, with waveforms like those of tectonic events and
405        irregular-shape frost tremors with frequencies of about 120-180 Hz.





4. The frost quakes sources in Talvikangas are mainly located on irrigated wetland cut by drainage channels while in Tähtelä about 50% of recorded frost quakes were caused by ice fracturing on the Kitinen river. However, several relatively strong events with origin both in irrigated and non-irrigated wetlands were also recorded there.

5. A significant number of sources of frost tremors are also confined to irrigated wetland areas and to roads cleaned from snow during winter, both in Talvikangas and in Tähtelä. In Tähtelä, a lot of tremors originated also from river ice.

6. The ground accelerations and ground velocities calculated for surface waves produced by the strongest frost quakes and recorded at epicentral distances of hundreds of meters from the source are noticeably large. Approximately, they correspond to V class and II-III class of Modified Mercalli Intensity Scale, respectively.

7. The high-frequency frost tremors corresponding to surface fracture opening in the uppermost frozen surface layer can directly damage infrastructure objects like roads and basements of buildings, which is confirmed by evidence of local inhabitans in Talvikangas.

8. Our study shows that frost quakes and frost tremors that occurs in the CZ during periods of seasonal freezing should be considered as phenomenon, that potentially can damage infrastructures. This has to be taken into account in seismic hazard assessment.

Our research is the first instrumental study of seismic events originated from wetland areas. It shows that these events can occur as a result of interaction between the uppermost layer of the solid Earth's CZ and atmosphere processes that deserves further study.

**Author contribution:** NA took a part in planning experiment, developed algorithm of detection tremors from continuous seismic recordings and calculate their source locations, processed seismic data, took part in interpretation of results and manuscript preparation. EK is a PI of ADAPTINFA project, she proposed the idea of the research and took a part in planning experiments, interpretation of results and writing the paper text. KM took part in project field work planning, organizing, implementation and data handling and manuscript writing. JO is a co-PI of the ADAPTINFA project that took part in planning of the experiment, field work and discussion of results. E-RK took part in planning experiment and field work.

**Competing interests**

The authors declare that they have no conflict of interest.

**Acknowledges**

This study is a part of the ADAPTINFA (URBAN ENVIRONMENT AND CLIMATE CHANGE IN THE ARCTIC: DATA-DRIVEN INTELLIGENCE APPROACH TO MULTIHAZARD MITIGATION) project funded by Academy of Finland in 2022-2024 (decision number 348802). Authors are very grateful to Sodankylä Geophysical Observatory (SGO) for providing



area for seismic experiment and assistance in conducting the measurements. The assistance of the SGO staff members Jyrki Manninen, Tero Raita, Toivo Iinatti and Alexander Kozlovsky is highly appreciated. We are very thankful also to people from
Talvikangas for sharing information about their observations of macro effects of frost quakes on 6.01.2023. The equipment of the FLEX-EPOS Finnish National Pool of mobile seismic instruments was used in recordings. The staff of the Institute of Seismology of the University of Helsinki assisted greatly with raw data processing and logistics during field campaign.

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
