# Peer review of "Frost quakes in wetlands in northern Finland during extreme winter weather conditions and related hazard to urban infrastructure."

_EGUsphere, 2023_

## Author Response (AR1)

Referee: General Comments

Overall, this is a good paper that requires only minor revisions. The paper documents two types of frost-related seismic activity: frost quakes and frost tremors. The methods are adequately described. The evidence from the timing of the events in relation to meteorological data and from the location of the events in relation to water and wetlands make a good case that the signals do indeed derive from the proposed freezing-related mechanisms. The recognition of this phenomenon is significant for the hazard that may pose to infrastructure and for understanding the sources of ambient seismic noise in the critical zone. I have only one major comment, regarding the use of HVSR to identify a frequency band of interest (see below), for which the explanation does not really make sense to me.

Specific Comments

Lines 117-124 : I am confused by this. Okkonen et al. (2020) Fig. 2B shows significant vertical as well as horizontal motion, and in the accompanying text they state that the signal is "depleted in body wave energy but having large-amplitude Rayleigh waves". If the signal is dominated by Rayleigh waves, shouldn't it have both horizontal and vertical motion and be elliptically polarized in a vertical plane?

Authors: In the case described in Okkonen et al. (2020) the frostquake was recorded only by single permanent OUL station, located at 14 km from the source. The source of the event appeared as a crack on the road of significant size (about 1m deep and half meter wide). The time of this event was reported by locals as the crack was formed exactly near their house. We found the record of this event on the seismogram of closest seismic OUL station (14 km from source) using this knowledge. On the same day, this station recorded huge number of impulse-like events. They appeared in OUL seismogram during this day only, when the air temperature significantly and abruptly dropped. This was unusual signal, as this station is located in a very quiet area on the bedrock. We analyzed the swarm of these impulses and found out similarity between waveforms of events from the swarm. We noticed from analysis of 3-component recordings that they mainly have horizontal polarization. Assuming that all of these impulses were produced by local sources (because they were presented only by surface waves), we suggested that the main source mechanisms of these events are vertical fracture opening. The recording of the strongest event, mentioned in Okkonen et al. (2020), had strong enough Rayleigh wave. But this is because of the relatively large distance from the source and the effect of propagation. In our pilot study in 2019-2020 as well as in the study described in the current paper, we analyse the events (tremors) with much smaller epicentral distances (less than 1000 m). Assuming that they should have the same polarization as impulses, detected in 2016, we used the HVSR as so-called "polarization filter" to detect frequency band of these events. This is possible because considered tremors look like a continuous swarm composed of events originating almost every second. Therefore, we could consider this process as continuous emission. Calculating the HVSR is working as averaging these impulses over time. Therefore, the HVSR maxima should correspond to the dominating frequency range of impulses.

Referee: From your results in this paper, it also looks like it is only the P waves that are horizontally polarized (Figure 6), and the Rayleigh waves as well as the high-frequency tremor in Figure 9 are elliptically polarized. I do agree that the change in HVSR at freezing temperatures is

significant, but it could also be explained by a change in the elastic properties of the subsurface due to ground freezing, rather than by a change in the noise source.

**Authors:** When we detected such HVSR behavior, the hypothesis about changing mechanical properties of soils was the first we tested. Practically, our first idea was to use the HVSR to understand changing in soils mechanical properties with time (namely, depth of the frozen layer). But when we calculated the time series of HVSR, we found significant disagreements with our original hypothesis. Firstly, the frequency band of HVSR maxima was too low. We had a soil station co-located with the seismic sensor during experiment. From this data we knew that the soil has been frozen only from ground surface down to 5 cm. In that case HVSR peak should be significantly higher (in order of 1kHz). In our case, the frequencies of 120-180 Hz would correspond to a layer of more than 1 m thickness. Moreover, the HVSR maxima in considered frequency range disappeared during about 6 hours after the air temperature became positive. The soil conditions could not be changed so fast. Before making conclusions, we checked the seismograms in the considered frequency range. These seismograms were quite similar to those we observed in 2016: a swarm of horizontally polarized impulses. This significantly supports origin of the HVSR maxima.

**Referee:** Instead of HVSR, or in addition to it, I would suggest plotting a spectrogram of the ambient noise over time. (This could be done with one or both of vertical and horizontal components, or an average of them.) If frost quakes and tremors are a major source of noise in this frequency band, then the same pattern should appear: The noise should occur only when the meteorological conditions are suitable, but the change will be more clearly related to the source, not the medium.

**Authors:** We analysed these spectrograms. But we did not detect such obvious correlation between spectrograms and air temperature. This is because of anthropogenic noise, as both sites are seismically not quiet. To find a way to reduce the effect of anthropogenic noise, we used the HVSR method, assuming the source mechanism as vertical opening and taking into account that we record seismograms in near-field area. We found correlation between air temperature and the HVSR behavior. To be sure that the frost tremors correspond to the frequency band detected by the HVSR, we tried to detect tremors in other frequency bands, but without success (they disappeared or no correlation with temperature was observed).

**Referee:** Technical Corrections

- The word "the" is sometimes missing. In the abstract I counted four instances: On, line 8 before "so-called," line 10 before "relationship," line 13 before "municipality," and line 24 before "strongest". The error doesn't appear to be as frequent in the rest of the manuscript, but it would be good to go through and check.

**Authors:** The text has been checked and corrected.

**Referee:** Line 26: Missing "and" before "cargo trains".

**Authors:** Corrected

**Referee:** Line 34: Hyphenate "snow-dominated"

**Authors:** Corrected

**Referee:** Line 40 "stats" should be "starts"

**Authors:** Corrected

**Referee:** Line 80: "is" missing before "located".

**Authors:** Corrected

**Referee:** The Figure 1 caption says "(Mention that units are in metres, X is Easting and Y is Northing)". This looks like some editing comment that wasn't fully implemented.

**Authors:** Corrected

**Referee:** Line 152: "moth" should be "month"

**Authors:** The typo has been corrected.

**Referee:** Line 167: "such significantly" should probably be "so significantly".

**Authors:** Corrected.

**Referee:** Figure 11 caption: The meaning of the blue dot is repeated (and "being" is misspelled in the first instance).

**Authors:** Corrected.

**Referee:** Line 325: The comma after latitude is not needed.

**Authors:** Corrected

**Referee:** Line 362: "shacking" should be "shaking".

**Authors:** Corrected

**Referee:** Line 366: "1-store buildings" should be "1-story buildings".

**Authors:** Corrected

**Referee:** Line 418: "phenomenon" should be "phenomena" since it is referring to two things.

**Authors:** Corrected

*Anonymous Referee #2*

**Referee:** This is an interesting paper on "frost quakes and tremor" detected in Finland.

I had never heard of such signals before.

The topic is interesting both for seismologists (as a new source of seismic signals), because of the associated damage and hazard and for researchers interested in the Cryosphere.

I recommend this manuscript for publication after minor modification.

I believe that several points should be clarified or improved but that the main results are correct.

Earlier experiment during winter 2019-2020 (l115-116).

Could you describe the experimental setting and the results?

**Authors:** The description has been added to the text.

**Referee:** l115-120, you wrote that "the frequency band of 120-180 Hz was selected based on results of our earlier pilot experiment in Talvikangas during winter 2019-2020.

In that experiment we recorded three component continuous seismic data by a single station in Talvikangas equipped by broadband Trillium compact seismometer, by Nanometrics Inc. (USA).

The results of our pilot frost quake study (Okkonen et al., 2020) suggest that the seismic signal excited by fracturing in the uppermost soils would have mainly horizontal polarization, as the main fracturing mechanism in that case is vertical fracture opening."

This paragraph (l115-120) is rather confusing for several reasons.

First, I did not find any description of the 2019-2020 experiment in the paper of Okkonen et al. (2020).

**Authors:** The paragraph had problems with formulation. From our observations described in Okkonen et. al., 2020, we assumed that the main source mechanism of considered events were vertical opening, as we observed a crack on the road and analysed polarization of impulses in the swarm on 6.01.2016, when the air temperature abruptly dropped and frostquake was reported by locals in Talvikangas. To check this assumption, we organized a pilot experiment in 2019-2020. We installed a broadband seismometer in Talvikangas and detected a swarm of impulses, but in another frequency band (120-180 Hz). They also had horizontal polarization. The swarm was looking like a continuous signal composed of events originating almost every second. That is why we decided to use the HVSR as co-called "polarization filter" to understand behavior of these swarms. Using this approach, we found some correlation with air temperature. Based on this, we assumed that these swarms are frost tremors, but we could not study origin of them, as we had records of only single station in our pilot experiment. When we obtained the data from our experiment in 2022-2023, we checked our hypothesis, by locating sources of signals in considered frequency band. We found that locations of sources correlate with wetlands as well as with reports by local inhabitants about new cracks and noise.

The text has been corrected.

**Referee:** Okkonen et al. (2020) describe data recorded in 2016 with a single sensor. It does not mention tremor.

**Authors:** The text has been changed. See previous comment.

**Referee:** It also assumes that these quakes were induced by fracture opening based on direct observations of cracks on buildings and roads, not from the analysis of seismic signals.

Is this reference wrong?

**Authors:** This reference is not wrong, but it is not describing all the details, related to seismic data analysis, as this work aimed to explain mechanism of frostquakes using numerical simulation.

**Referee:** Also, the polarisation of the seismic signal depends not only on the source mechanism, but also on the type of waves (P,S, Rayleigh...) and on the source azimuth.

This is explained latter (l195), so that the discussion on the signal polarization could be removed from section 2.

**Authors:** That is correct, but in section 2 we explain our assumptions which have been done from direct observations and results of our pilot experiment in 2019 (HVSR). The features of the signals, which we found in 2023 are partially proving our assumptions.

**Referee:** The frequency band of 120-180 Hz mentioned on l115 corresponds to the tremor signal but the quakes have a lower frequency content.

Could you please clarify this part?

**Authors:** Because the sources of these events are of different scales. In our paper we did not aim to explain everything related to these events. We just detected, located them and estimated ground motions produced by them. We found that only events in frequency bands 2-20 Hz and 120-180 Hz have correlation with air temperature and originating during these extreme weather events.

**Referee:** Figure 2: the largest peak of seismicity occurred on 2016/2/10 but is not discussed in the text.

Do you know what was the source of these events? Are the signals similar to frost quakes?

**Authors:** They are like impulses from swarm, recorded 6.01.2016. We could not check their origin, because we had records of only single permanent OUL station. We used detector based on correlation analysis to detect them and we just can say that waveforms of these events look quite similar to thoss of events from swarm, recorded in 6.01.2016. However, these events in 2016 motivated us for more detailed instrumental study in 2022-2023, which we report in our present paper.

**Referee:** Figure 3b.

Nice tremor signal with a clear correlation with temperature data!

How do you interpret the lower frequency 50 Hz signal when the high-frequency tremor stopped?

**Authors:** We investigated the seismograms in the frequency range near 50 Hz as well, but did not find a correlation between air temperature and this maxima behavior. Probably the origin of this maxima is not related to considered phenomena.

**Referee:** Could you show the spectrogram of the signal for the full duration of this 2019-2010 experiment and a zoom on the time window when tremor was detected?

Could you also show a spectrogram for the 2022-2023 experiment for comparison?

**Authors:** We analysed these spectrograms, but they are not very informative, as both sites are noisy (urban areas). We did not find clear correlation between spectrograms and air temperature. This was one of the reasons why we used the HVSR diagrams instead of spectrograms.

**Referee:** Source mechanism

Did you try to estimate P wave polarity to constrain the source mechanism?

**Authors**: We did not analyse records in such details. The main goal of current paper is to report about a new type of seismicity we observed as well as about possible hazard, related to this seismicity. We will do detailed analysis of seismograms in our further research, using the whole dataset of our experiment.

**Referee:** Vertical motion should be up on all sensors for crack opening. Is it what you observed?

**Authors:** We did not analyse records in such details. See previous comment.

**Referee:** Could you also estimate magnitude (local or moment magnitude)?

**Authors:** In our opinion, ground velocities and accelerations are more informative for seismic hazard assessment in considered cases, as we recorede events in near-field area. In our further study, we are planning to estimate magnitudes as well as to study source mechanisms in more detail.

**Referee:** Temporal evolution of tremor and quakes

Did you observe tremor at both sites?

**Authors:** Yes (see figure 12)

**Referee:** Fig 13 shows the "number of seismic events": does it includes both quakes and tremors?

**Authors:** It means number of frost quakes. Fig. 13 caption has been corrected.

**Referee:** Could you show a figure with the number of events per hour for each class (tremor, quakes) and the temperature at each site?

**Authors:** A sub plot, that illustrates number of frost tremors per hour has been added to figure 13.

In our paper we analysed a single frost quakes episode during 6.01.2023, aiming mainly to locate events and to evaluate ground motion produced by them. As our previous study shows (Okkonen et al., 2020), the air temperature is important factor that could initiate episodes of frost quakes, but it is not the only one. Also snow depth and soil moisture content are important factors. In our next paper, which is now under preparation, we are analysing the whole winter period and the number of events not only for freezing, but also for melting periods. In the next study we take into account not only temperature variations, but also those other factors. The paper will include also detailed thermomechanical modelling. It is not possible to include everything into one single paper.

**Referee:** Quake detection

l190. What do you mean by "seismograms were ... stacked using beam forming algorithm"?

Could you provide a reference or describe the beamforming method you used?

Is it a simple stack (sum) or do you shift the signals in time? How is this time shift estimated?

Does it really improve the signal quality and the location accuracy compared to picking the raw signals?

**Authors:** We used a method described in Schweitzer et al., 2012 and the reference has been added to the text. We shifted seismograms in time, according to recommendations given there. This procedure helped to improve signal-to-noise ratio and we used these stacked seismograms only on visual inspection (detection) stage. To calculate source location, ground velocities and accelerations, we used seismograms recorded at each sensor.

**Referee:** Figure 6: What are the signal shown in (a): the stack for each array or individual signals? Did you filter the signals?

Same questions for Fig 8.

**Authors:** Figure 6 shows seismograms, stacked for each array using beamforming, mentioned above. The same is for Figure 8. There are averaged ground velocities and accelerations over the array.

**Referee:** Figure 7. Could you add the sensors on these figures?

**Authors:** Done

**Referee:** Tremor location

The description of the location method is a bit confusing.

Could you explain what signal you correlate? I think you correlate each sensor with the central sensor of the same array but it should be written more explicitly.

**Authors:** That is correct. But this was not always the central sensor. To avoid errors in detection because of low quality of records of central sensor, our algorithm randomly selects several candidates to "virtual source" and tests them. If correlation coefficients are always lower than the threshold (for all candidates to "virtual source"), then the algorithm reports no events in the interval considered. Otherwise, the seismic event is detected and the sensor with the highest correlation coefficient is selected as a "virtual source".

**Referee:** The term "virtual source" is also misleading. If I understood correctly this is the central sensor of each array?

**Authors:** This is not always the central sensor, but those with which all other records correlate (the first record in cross-correlation procedure). We are selecting this sensor by tests. If low correlation is detected between all sensors and the central one, the algorithm selects randomly another one as a possible "virtual source" (see previous comment). The term "virtual source" is commonly used in seismic interferometry. We think that usage of this term is justified in our case, as we are using interferometric approach too. We transform seismograms in such a way that after this transformation they look like impulse response of the source, located in point with the "virtual source" cooordinates.

**Referee:** Did you also try to use this method to locate frost quakes?

**Authors:** We tried, but in our study we preferred to use visual inspection of seismograms and a manual picking of arrivals, as a more reliable method, because we deal with unique data showing quite new phenomenon.

---

## Author Response (AR2)

**Referee:** The explanation of the HVSR makes more sense now. If I understand correctly, the previously detected tremor had a distinctive high HVSR, so this distinctive feature was used to find other instances of frost tremor. That makes sense regardless of the reasons for the high HVSR. Furthermore, the authors' explanation of why the HVSR change is inconsistent with a change in the subsurface medium makes sense to me.

However, I am still confused by the elliptical polarization shown in Figure 9, which is identified as being typical of tremor. This figure appears to show larger vertical than horizontal amplitude. That would be inconsistent with the high HVSR and horizontal polarization indicated by Figure 3 and the associated text. Can you clarify this apparent discrepancy?

**Authors:** In the pilot experiment in 2019 we made observations using a single 3-component station. We just found such long-term trend in HVSR behaviour and dependence on temperature and this motivated us to make a new experiment. However, Figures 3 and 9 cannot be compared directly, as they correspond to different time scale. In Figure 3 we show a general trend in HVSR during a long time period, while in Figure 9 we show just a single example of frost tremor from the swarm composed of overlapping events. In that particular case the polarization is vertical, indicating that the event is close to the station. This polarisation can correspond to P-wave (see also the Figure 6 that shows that near the source we record mainly body waves). Such polarization in the near-field zone for this particular event is in agreement with evaluation of radiation pattern from the surface tensile fracture opening in ice (Dudko, 1999), although other modes of fracturing are also possible. Concerning the observed HVSR maxima at high frequences in Fig.3, it shows the HVSR calculated from overlapping signals from multiple sources arriving to the station from different distances. In the case of relatively large distances, the main energy of the signal would corresponds to surface waves and the energy from these sources is dominating in the signal recorded at this particular station. This is in agreement with our present result, in which we found multiple events originating from wetland located at distance of 300 m from the site of our 2019 station. However, we are not stating that the high HVSR in Fig. 3 is purely due to frost tremors, as it would be too much for interpretation of single station observations. The explanation proposed in Dal Moro (2020) that the HVSR is increased due to appearance of higher mode Love waves cannot be excluded. We have modified the text to clarify the difference in polarisation (lines 119-134 and 336-348)

**Referee:** Finally, in the text added on line 125 of the manuscript with tracked changes, you refer to "winter 2019-2019". Should this say "winter 2019-2020"?

**Authors:** The typo has been corrected

**Anonymous referee #2**

**Referee:** The authors have answered most questions raised by the other reviewer and myself, but they did only minor changes in the manuscript.
I think that questions raised by the reviewers maybe be the same as interrogations of the readers, and many answers should be included in the manuscript.

Even negative results (eg, absence of correlation with temperature) should be included. For instance, I really think that showing a spectrogram of the signal is interesting, at least as supplementary material.

**Authors:** We have calculated spectrograms at the initial stage of our study, when we processed the data of our pilot experiment in 2019, but they really are not very informative, as the site in Talvikangas is located in urban area. So the specrograms are just dominated by typical diurnal variations of anthropogenic noise obereved at numerous urban sites around the globe (see, for example, Steimann et al., 2021) and no any distinct correlation with temperature are seen there. The correlation with temperature we noticed only at the HVSR plot, that is why we decided to include namely this result into the manuscript, in order to avoid overloading the text with non-informative figures. We would like to notice that the main scope of the paper was not to study seasonal variations of properties of the subsurface, but the fracturing in the subsurface during short-term rapid air temperature drop and associated hazard. We added some explanation concerning this to the text (lines 128-135)

**Referee:** The authors did not answer some suggestions: eg, looking at direction of first movement or computing magnitudes, saying that this is left for future work.

**Authors:** During our work on the manuscript we had to make some prioritisation what to include into the manuscript not to make it too long. That is why we decided to concentrate mainly on events locations and classification and on evaluation of associated seismic hazard that can be evaluated directly in terms of ground accelerations recorded in the near-field zone. By this, and by selection of the Cryosphere journal, we were trying to broaden our target auditorium and to make our paper interesting also for specialists studying extreme weather conditions (like sudden cold waves of various scales) and their effect on urban infrastructures and for geotechnical engineers dealing with operation of these infrastructures in cold climate. The latter specialists are mainly used to work with ground accelerations and ground velocities. This was also motivation why we decided to consider the episode of 6.01.2023 when extreme rapid temperature drop was observed simultaneously at two of our sites located at distance of about 250 km (see Fig. 5). Another motivation were concerns of local people in Talvikangas, who heared the noises during that period and noticed direct damages to roads etc.
We definitely will make analysis of records of frost quakes and frost tremors in details (including magnitudes, the first motions for those events where they could be analysed) for the whole experiment period and prepare more seismologically oriented paper that we are going to submit to the more seismologically oriented journal. But this need to be a separate work. The current mansuscript aims to report our observations about new type of seismicity originating from wetlands and also to evaluate associated seismic hazard.

**Referee:** What I just realized reading the revised manuscript, is that they recorded 6 months of data but only analyzed one day of data (2023/1/6) corresponding to the main temperature drop.
The analysis of the full dataset is left for future work. I personally don't like this practice of cutting work in many papers.
The authors did a good job at identifying different types of events (quakes and tremor) and locating them.
But the problem is that the origin of these events is not clearly established.
With only one day of data it is hard to establish a correlation between quake events and temperature drop.

They have shown that many events are detected on that day (2023/1/6), not that this seismic activity was particularly large.
The authors need to be much more cautious when interpreting the source of these events, or better, to analyze the data for a much longer time period.

**Authors:** We partly answered to this comments in our reply to the previous question. Our aim was to consider the particular extreme weather episode of rapid air temperature drop. The episode was of regional scale, as the same process was observed at two our observation sites (see Fig.5). In our Fig. 13 it is seen that the events appeared namely during the period of rapid temperature drop. However, our previous study (Okkonen et al., 2020) suggests that there is no direct correlation between extreme low air temperatures and large activity of frost seismic events, as the latter is depending on thermal stress in uppermost subsurface that, in turn, is depending not only on air temperature variability, but also on other factors (in particular, on snow depths and soil moisture, which are varying during winter period in our geographical area). In our next paper, which is now under preparation, we shall present results of thermal stress modelling for the whole observation period, but we prefer to publish details of seismic signal analysis and events detection algorithms in a separate paper. It is really not possible to put everything in a single paper.

**Referee:**
Details comments

The quake detection method is not described.
Signals are stacked on all sensors of the network, but how are these stacked signals used to identify quakes?

**Authors:** We processed the data manually, using visual inspection of seismograms. After location of some events we noticed that they are originating from wetlands and analysis of these interesting events cannot be just delegated to authomatic detection algorithms. We were picking arrivals manually and calculated source locations, using information about P and S waves arrivals and velocity model shown in Fig. 4a. In addition, visual inspection of seismograms helped us to recognise tremor-type signals and to develop an algorithm for their detection.

**Referee:** Rate of tremor events
The detection method is not clear. You analyze a time window of 0.8 s, but what is the time shift between successive windows?
Is there some overlap? If so, how do you remove duplicate events?

**Authors:** We used time shift of 0.3 s and 10% overlap. In case when events has been detected, we shifted time window to full its lengt (0.8 s) to avoid "double detection" of the same event. The correspondent text is added.

**Referee:** Location of tremor and quakes
The location of quakes is shown in Fig 7 and tremor are shown in fig 12.
It would be good to merge these figures in one pin order to allow a comparison of the location of quakes and tremor.

**Authors:** We illustrated these results on different figures to avoid overloading of a single figure with information. For us it was also important to show location of roads and irrigation

channels. If all the sources (tremors and events) are in the same plot, some of source locations of tremors can be invisible in that case.

**Referee:** Polarization
I still don't understand the sentence in section 2.
"The results of our earlier observations (Okkonen et al., 2020) suggest that the seismic signal excited by fracturing in the uppermost soils would have mainly horizontal polarization, as the main fracturing mechanism in that case is vertical fracture opening. "
Do you consider only P waves? Could you add a reference?
Your figures 6 (quakes) and 9 (tremor) show on the opposite larger amplitudes for the vertical components.
Could you clarify this point?

**Authors:** In our original study of frost quakes swarm in 2016 (Okkonen et al., 2020) we had only recordings of frost events by single permanent station OUL located at distance of 14 km from the area in Talvikangas, where multiple fractures at the surface were formed as a result of vertical fracture opening (so we were just seen these fractures on the sutface). At such distance (e.g. in far-field zone) the main signal that could be identified was from surface waves, and body waves were not visible in seismograms. These "ground-truth events" and the waveforms recorded by OUL stations were the background of our suggestion that we need to search the waveforms with similar polarization. When processing the data of our experiment in 2022-2023, we found also body waves, in particular in near-field area that have different polarisation (see Fig. 6). See also our reply to the comment of the Reviewer #1 concerning different polarization.
We agree that this sentence is misleading and we modified that part of the text

**Referee:**
Source of tremor
In section 4.2 you write that "Comparison of waveforms of such events with that of the 245 single frost quake described in Okkonen et al. (2020) suggests the similar source mechanism, namely, vertical fracture opening"
Could you give more details? Which characteristic do you consider?

**Authors:** We used shapes of these signals. One of the features we used is absence of clear P and S arrivals as well as the higher horizontal amplitude than the vertical. Of course that is not always true, as we found later, analysing tremors (figure 9). But it was our original hypotesis, based on our pilot studies. We used this when experiment planning and on the first steps of data processing.

**Referee:** The fact that the signal has an elliptical polarization suggests that it is dominated by Raleigh waves and thus is rather shallow.
But why do you suggest that the source is vertical fracture opening?

**Authors:** This is our original hypothesis, made based on our previous studies and from direct observation of fractures on the surface. Dominated Raylegh wave might be because of propagation effect. We think that our hypotethis was reasonable taking into account that we had only single station records in our pilot experiment. This hypothesis proved only partially in the research, described in the current manuscript and we modified the text to explain this.

**Referee:** Fig 9 shows 30s of seismic signal. One tremor event is highlighted. Is this the only event detected during this time window? This is surprising because this does not correspond to peak amplitude!

**Authors:** In this figure we show a typical swarm of tremors and we highlighed the parts of records, which we used for particle motion diagrams. This is just an example. There was no peak amlitude, but high correlation coeffcient (see description of our location algorithm). Practically tremors usually have been recorded as swarms of overalapping single impulses, but not like a single event.

**References**

Dal Moro G., 2020. The magnifying effect of a thin shallow stiff layer on Love waves as revealed by multi-component analysis of surface waves. Scientific Reports, 2020 Jun 3, 10(1), 9071.

Dudko YV. Analysis of seismo-acoustic emission from ice fracturing events during SIMI'94 (Doctoral dissertation, Massachusetts Institute of Technology), 1999.

Steinmann, R., Hadziioannou, C., Larose, E., 2021. Effect of centimetric freezing of the near subsurface on Rayleigh and Love wave velocity in ambient seismic noise correlations, Geophysical Journal International, Volume 224, Issue 1, January 2021, Pages 626–636, https://doi.org/10.1093/gji/ggaa406.